# Growth Rhythm Analysis of Young Stand and Selection of Superior Families in *Choerospondias axillaris*

**Guangyou Li** [1], **Jianmin Xu** [1], **Juan Li** [1], **Canzhang Lu** [2], **Haifei Lu** [1], **Baoying Mai** [2], **Mingdao Luo** [2] **and Chunjie Fan** [1,*]

[1] Key Laboratory of State Forestry and Grassland Administration on Tropical Forestry, Research Institute of Tropical Forestry, Chinese Academy of Forestry, Guangzhou 510520, China
[2] Forestry Research Institute of Gaoming, Foshan City, Foshan 528515, China
[*] Correspondence: fanchunjie@caf.ac.cn; Tel.: +86-139-2875-8310

**Abstract:** As an important source of medicine, food, and high-quality wood, *Choerospondias axillaris* has been widely planted in South China. However, few studies of its growth traits and rhythm and concerning the selection of superior provenances/families were developed. In this study, 77 families from five provenances were studied and grouped. Tree height, diameter at breast height (DBH), and crown width within and between rows were measured. The relevance between growth traits and environmental factors was also analyzed. The results showed the height, DBH, and crown width within and between rows were divided into slow, mean, and fast growth periods, which was significantly related with sunshine hours in earlier months. Next, monthly rainfall showed an extremely positive correlation with the increase in tree height, DBH, and plant volume. Then, the monthly volume increment had a significantly positive correlation with five meteorological factors, excluding sunshine hours with increasing months, while the monthly growth of tree height and crown width within and between rows was significantly negatively correlated with sunshine hours. Based on these results, the superior families No. 15, 76, and 56, which originated from the Raoping provenance, were selected for their fast growth and available adaptation. These results provided the reliable growth rhythm of a young *C. axillaris* stand, which established a basis for fertilizing and forest tending. During these processes, meteorological factors, especially humidity and sunshine duration, had important effects on growth, implying that the external climate should be paid more attention to promote fast growth and prevent diseases and insect pests. Moreover, selecting superior families was helpful to further breeding and plantation.

**Keywords:** *Choerospondias axillaris*; family; plantation; meteorological factors; correlation

## 1. Introduction

*Choerospondias axillaris* (Roxb.) Burtt et Hill is famous for its fruit, used in wild jujube cake. It is widely grown and planted in South China, and grows on hills, hillsides, or in valleys with an altitude of 300–2000 m and is suitable for soil with deep fertility and good drainage, especially in light, warm, and humid weather [1,2]. It is a popular tree species for landscaping [3] with a high wood quality used for mahogany furniture; it is also a source of plant medicine [4–7]. Meanwhile, extensive adaptability means it is easily grown in acid, neutral, or calcareous soil. Moreover, it can regenerate naturally and has strong germination ability while possessing a low occurrence level of diseases and insect pests. Additionally, *C. axillaris* has stronger fast-growing ability and a good afforestation effect after artificial cultivation, with broad development prospects in China for forest classification management [8–10].

Growth rhythm is the growth status and trend in a plant in a certain period of time in specific environment, including the beginning and ending time of growth, the length of the growth period, and the speed of the growth rate, which is closely related to the

plant's size during growth [11]. The study of tree growth rhythm [12–15] can reveal the growth characteristics of an early artificial cultivation period, provide theoretical basis for the management of target trees, and also help to introduce and evaluate germplasm resources, and accelerate the promotion of suitable germplasm in suitable areas as far as possible [16]. Research on the growth rhythm of multiple tree species, such as *Larix gmelinii* (Rupr.) Kuzen, *Picea abies* mast, *Tapiscia sinensis* Oliv., *Salix matsudana* Pendula, and *Zenia insignis* Chun, has been developed, including at the seedling and young forest stages [17–22]. Ground diameter growth and seedling height conform to the S-shaped growth curve and seedling height and ground diameter are divided into different growth stages [23,24]. During the growth stage, strengthening water and fertilizer management is an important measure to save water, fertilizer, and material resources to achieve a higher input/output ratio.

Meanwhile, growth rhythm can help us when selecting superior families and clones by early growth. Additionally, developing the early selection and genetic variation analysis in tree species was necessary to ascertain superior families and clones and further improve the output value of trees such as *Eucalyptus saligna*, *Cunninghamia lanceolata* (Lamb.) Hook., *Cryptomeria japonica*, and *Pinus contorta* [25–28].

Moreover, various tree species have their own fast-growing periods in a year, which are related to genetic correlation and long-term climate. Regarding long-term adaptation to the external climate, the growth rhythm differs among tree species such as *Hevea brasiliensis* (Willd. Ex A. Juss.) Muell., *Quercus rubra* L., and *Pinus kesiya* Royle ex Gordon [29–31]. This indicates that meteorological factors have a certain impact on growth [32,33]. The correlation of meteorological factors and growth rhythm can be a useful supplement when selecting superior and stable genotypes [34].

However, there are few studies on the growth characteristics and growth rhythm of the *C. axillaris* plantations, although some studies on artificial *C. axillaris* cultivation have been performed. Meanwhile, none of the superior provenances/families suitable for the current planting environment have been selected until now, which limited the large-scale application of *C. axillaris* plantation. To improve the growth rate and further expand the plantation area, it is urgent to know the growth rhythm of various families and further select superior provenances/families with fast growth and wide adaptability. Hence, 77 families of *C. axillaris* were used to assess their growth rhythm and select appropriate families/superior plants, which will provide a theoretical basis for young forest tending and management.

## 2. Materials and Methods

### 2.1. Experimental Setting Area

The experiment was performed in Forestry Research Institute of Gaoming District, Foshan, which is located southwest of Foshan City, Yangmei Jiling (22°47′36″ N, 112°41′48″ E) (Figure 1) of Yanghe Town. The annual average temperature is 22.5 °C and the annual rainfall is 1681.2 mm. The forest land in the planting area was mainly red soil in mountainous area, with an average altitude of 50 m, soil organic matter content of 17.007 g kg$^{-1}$, available N, P, and K contents of 89.01, 0.839, and 55.98 g kg$^{-1}$, respectively, and available B and Zn of 0.165 and 1.000 mg kg$^{-1}$, respectively, with pH 4.41 (soil:water = 1:2.5). The experimental area's grown plant species included *Ilex asprella* (Hook and Arn.) Champ. ex Benth., *Desmos chinensis* Lour., *Mallotus apelta* (Lour.) Muell. Arg., and other common herbs, such as *Gleichenia linearis* Clarke and ferns (*Adiantum* spp.), among others, before plantation. In this study, a randomized complete block design was adopted, with 77 families, 5 plots, and 8 replicates. Planting holes were 50 × 50 × 40 cm (length × width × height).

Five main meteorological factors including temperature, rainfall, sunshine hours, and humidity were recorded and calculated. As shown in Table 1, the daily average maximum temperature reached the highest from June to July, daily average and minimum air temperature showed the lowest from December to February. The rainfall level was high

from May to June and low from November to January, and humidity was high from March to May and low from January to February. The details are listed in Table 1.

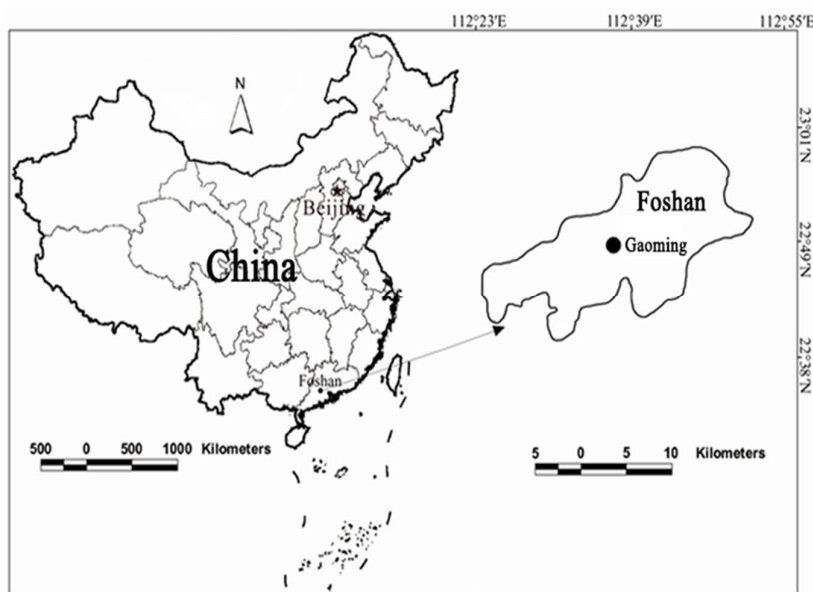

**Figure 1.** Location of study sites (black circles) in Gaoming of Foshan City, Guangdong, China.

**Table 1.** The effect of five meteorological factors in the period of investigating.

| Growth Survey Month | Average Daily Temperature (°C) | Daily Maximum Temperature (°C) | Daily Minimum Temperature (°C) | Rainfall (mm) | Sunshine Hours (h) | Daily Air Humidity (%) |
|---|---|---|---|---|---|---|
| 2017.7 | 28.05 | 36.20 | 23.30 | 151.80 | 145.50 | 83.17 |
| 2017.8 | 28.13 | 38.40 | 21.70 | 260.40 | 230.70 | 81.67 |
| 2017.9 | 27.48 | 36.20 | 25.40 | 25.00 | 182.20 | 85.08 |
| 2017.10 | 23.31 | 33.80 | 24.30 | 52.40 | 216.10 | 78.56 |
| 2017.11 | 18.99 | 28.70 | 20.90 | 42.70 | 79.20 | 81.77 |
| 2017.12 | 14.71 | 24.10 | 15.30 | 0.10 | 180.60 | 66.28 |
| 2018.1 | 13.19 | 25.00 | 17.00 | 57.8 | 100.10 | 82.13 |
| 2018.2 | 13.80 | 29.00 | 16.90 | 4.20 | 112.40 | 76.46 |
| 2018.3 | 13.80 | 29.00 | 1.60 | 8.60 | 112.40 | 76.46 |
| 2018.4 | 19.23 | 30.10 | 5.30 | 49.40 | 124.70 | 82.62 |
| 2018.5 | 21.59 | 31.40 | 7.90 | 53.60 | 70.60 | 84.32 |
| 2018.6 | 27.29 | 32.42 | 24.30 | 39.19 | 4.78 | 27.29 |
| 2018.7 | 27.29 | 35.70 | 21.80 | 587.80 | 143.40 | 85.23 |
| 2018.8 | 27.28 | 36.20 | 22.90 | 213.40 | 181.90 | 87.96 |
| 2018.9 | 26.30 | 35.70 | 25.20 | 153.80 | 165.30 | 86.02 |
| 2018.10 | 22.15 | 32.30 | 21.70 | 23.70 | 147.50 | 80.89 |
| 2018.11 | 19.67 | 29.50 | 20.40 | 16.90 | 131.40 | 85.88 |
| 2018.12 | 14.62 | 28.40 | 18.70 | 6.50 | 70.50 | 85.56 |
| 2019.1 | 14.62 | 28.40 | 5.20 | 11.20 | 70.50 | 85.56 |
| 2019.2 | 14.15 | 26.20 | 5.40 | 1.60 | 105.50 | 84.34 |
| 2019.3 | 17.62 | 29.30 | 8.80 | 85.00 | 51.40 | 88.86 |
| 2019.4 | 18.98 | 29.80 | 10.90 | 140.90 | 57.60 | 90.97 |
| 2019.5 | 23.52 | 33.20 | 15.80 | 292.80 | 71.80 | 92.97 |
| 2019.6 | 24.62 | 35.20 | 16.80 | 379.50 | 67.70 | 91.27 |
| 2019.7 | 27.70 | 36.10 | 21.40 | 339.10 | 143.70 | 88.60 |
| 2019.8 | 27.94 | 37.90 | 23.50 | 237.70 | 156.80 | 88.28 |
| 2019.9 | 27.77 | 37.50 | 23.20 | 350.40 | 180.90 | 88.50 |
| 2019.10 | 25.92 | 35.50 | 16.90 | 145.50 | 239.80 | 83.02 |
| 2019.11 | 23.71 | 34.80 | 15.10 | 27.10 | 203.60 | 80.11 |
| 2019.12 | 19.21 | 30.50 | 11.30 | 0.00 | 255.90 | 75.14 |

### 2.2. Experimental Design and Materials

The materials of 77 families originated from five provenances, including Anhua, Longchuan, Raoping, Lianshan, and Quanzhou, and the details are listed in Table 2.

**Table 2.** The provenances and families sourced in the trial.

| Family Codes | Quantity of Family | Provenance | Latitude (N) | Longitude (E) | Altitude (m) |
|---|---|---|---|---|---|
| 1–4 | 4 | Hunan Anhua | 28°24′ | 111°12′ | 650 |
| 5–9 | 5 | Guangdong Longchuan | 24°12′ | 115°18′ | 500 |
| 10–40, 52–77 | 57 | Guangdong Raoping | 23°43′ | 116°59′ | 30 |
| 41–47 | 7 | Guangdong Lianshan | 24°35′ | 112°05′ | 360 |
| 48–51 | 4 | Guangxi Quanzhou | 25°55′ | 112°05′ | 200 |

For the determination of growth rhythm, all 77 families were selected (see Table 1), and the growth was observed in five trees of each family for no less than one year (a total of 18 observations were conducted to calculate monthly growth increments), including tree height, diameter at breast height (DBH), crown width in and between rows, and growth traits.

The construction of test forest happened in July 2016 and there was an 89.09% preservation rate 3 years after planting, which was enough to develop the determination of growth rhythm.

The tree height (m), diameter at breast height (DBH, mm), crown width (m) within and between rows, growth traits, and the individual tree volume ($cm^3$) were measured and calculated each month. Meanwhile, daily meteorological data were collected from 1 June 2017 to 31 December 2019 at meteorological stations.

### 2.3. Data Analysis

To investigate the correlation of growth characteristics and growth rhythm of 3-year-old *C. axillaris* forests with meteorological factors including temperature, rainfall (R), sunshine hours (SH), and air humidity (HU), Excel and SAS software were used for analyzing and calculating. The single plant volume ($cm^3$) [35] was calculated by using formula:

$$V = H \times DBH^2 \times F \tag{1}$$

where H, DBH, and F represent tree height (m), DBH (mm), and rate of tree form, respectively. In this study, F value was 0.4.

Linear model was used for analyzing variance of unbalanced data [36]. SAS software version 8.0 [37] was used to analyze the variance of repeated single plant data of all the tested families at 3 years old, with the Duncan test assessing result significance. Meanwhile, SAS software was also used to analyze the Pearson correlation between growth rhythm and meteorological factors.

### 3. Results

### 3.1. Growth Traits Showed Significant Differences in Various Families and Superior Families Selection

Five growth traits, including H, DBH, V, CT, and CR showed significant differences in various families with the average values of 3.66 m, 44.34 mm, 3552 $cm^3$, 2.87 m, and 2.91 m, respectively (Table 3). According to their growth and growth rhythm of variance analysis, 77 families were divided into three groups. The first group (Cluster I) included 11 families, which were 15, 76, 56, 49, 17, 5, 48, 18, 71, 12, and 62; these were classified as a fast-growing type with a single plant volume range of 4872–5763 $cm^3$. It was found that all growth traits of the first group were 8.95% higher than the overall average of 77 families, in which the DBH, the single plant volume, the crown width between plants, the crown width between rows was 14.10%, 46.82%, 7.55%, and 8.61% higher than the overall average, respectively.

**Table 3.** Growth performance of *C. axillaris* families at 3 years old.

| Families | Height (m) | DBH (mm) | Mean Volume (cm³·tree⁻¹) | Crown Width in Rows (m) | Crown Width between Rows (m) |
|---|---|---|---|---|---|
| 15 | 3.97 ± 1.43 ABCDEFG | 50.21 ± 21.34 ABCDE | 5763 ± 64 A | 2.88 ± 1.05 ABCDE | 2.93 ± 1.05 ABCDEF |
| 76 | 4.03 ± 1.18 ABCDEF | 52.07 ± 19.43 ABC | 5720 ± 57 AB | 3.12 ± 0.89 ABCD | 3.16 ± 0.70 ABCDEF |
| 56 | 4.33 ± 0.29 A | 55.98 ± 9.94 A | 5459 ± 16 ABC | 3.27 ± 0.50 ABC | 3.47 ± 0.76 AB |
| 49 | 4.13 ± 0.94 ABC | 53.09 ± 14.83 AB | 5440 ± 40 ABCD | 3.16 ± 0.65 ABCD | 3.30 ± 0.73 ABCD |
| 17 | 3.88 ± 1.16 ABCDEFG | 51.20 ± 19.56 ABCD | 5380 ± 49 ABCDE | 3.36 ± 1.22 AB | 3.38 ± 0.98 ABC |
| 5 | 3.91 ± 1.16 ABCDEFG | 47.77 ± 20.3 ABCDEFG | 5170 ± 66 ABCDEF | 2.97 ± 0.53 ABCDE | 3.16 ± 0.67 ABCDEF |
| 48 | 4.02 ± 1.31 ABCDEF | 48.51 ± 18.07 ABCDEF | 5109 ± 55 ABCDEFG | 3.14 ± 0.94 ABCD | 3.19 ± 0.90 ABCDEF |
| 18 | 4.08 ± 1.00 ABCD | 50.24 ± 14.60 ABCDE | 4958 ± 47 ABCDEFG | 3.02 ± 0.71 ABCDE | 3.04 ± 0.85 ABCDEF |
| 71 | 3.88 ± 0.93 ABCDEFG | 50.28 ± 17.61 ABCDE | 4942 ± 39 ABCDEFG | 2.93 ± 0.92 ABCDE | 2.94 ± 0.94 ABCDEF |
| 12 | 4.06 ± 0.89 ABCDE | 50.69 ± 14.39 ABCDE | 4909 ± 33 ABCDEFG | 3.17 ± 0.67 ABCD | 3.27 ± 0.79 ABCDE |
| 62 | 3.67 ± 1.43 BCDEFGHIJ | 50.97 ± 15.94 ABCD | 4872 ± 42 ABCDEFG | 3.02 ± 0.79 ABCDE | 3.00 ± 1.00 ABCDEF |
| ⋮ | ⋮ | ⋮ | ⋮ | ⋮ | ⋮ |
| 4 | 3.15 ± 0.55 IJ | 37.23 ± 10.32 GF | 1980 ± 11 H | 2.56 ± 0.73 DE | 2.51 ± 0.70 F |
| 75 | 3.19 ± 0.41 HIJ | 36.16 ± 9.50 GF | 1856 ± 26 H | 2.60 ± 0.34 CDE | 2.56 ± 0.51 EF |
| 29 | 3.11 ± 0.50 J | 35.80 ± 11.52 G | 1854 ± 13 H | 2.37 ± 0.73 E | 2.53 ± 0.77 F |
| Mean | 3.66 ± 0.84 | 44.34 ± 14.65 | 3552 ± 3304 | 2.87 ± 0.79 | 2.91 ± 0.81 |
| *F* value | 2.78 | 2.32 | 2.74 | 1.99 | 2.33 |

Note: The same letter within a column indicates significant difference between means for each trait at $p < 0.01$ using Duncan's test.

The secondary group (Cluster II) contained 17 families including 61, 69, 37, 21, 13, 60, 27, 38, 70, 57, 19, 53, 26, 58, 4, 75, and 29, which were classified as a slow-growing type, with a single plant volume range of 1854–2837 cm³. The rest of the families were classified as Cluster III, belonging to the medium growth rate type, with individual plant volume ranging from 2855 to 4562 cm³ (Table 4).

**Table 4.** Classification of *C. axillaris* families based on growth performance.

| Types | Family Codes | Height (m·a⁻¹) | DBH (mm·a⁻¹) | Mean Volume (cm³·a⁻¹·tree⁻¹) | Crown Width in Rows (m·a⁻¹) | Crown Width between Rows (m·a⁻¹) |
|---|---|---|---|---|---|---|
| I | 15, 76, 56, 49, 17, 5, 48, 18, 71, 12, 62 | 1.33 ± 0.36 | 16.87 ± 5.74 | 5247 ± 324 | 1.03 ± 0.28 | 1.05 ± 0.29 |
| II | 73, 24, 11, 9, 46, 63, 52, 10, 45, 42, 47, 68, 36, 50, 67, 32, 25, 51, 41, 16, 20, 64, 31, 65, 55, 59, 54, 40, 44, 35, 33, 28, 2, 30, 6, 66, 74, 34, 22, 43, 7, 8, 23, 72, 14, 1, 39, 77, 3 | 1.23 ± 0.27 | 14.81 ± 4.88 | 3540 ± 49 | 0.96 ± 0.27 | 0.97 ± 0.28 |
| III | 61, 69, 37, 21, 13, 60, 27, 38, 70, 57, 19, 53, 26, 58, 4, 75, 29 | 1.15 ± 0.21 | 13.50 ± 3.86 | 2532 ± 339 | 0.91 ± 0.23 | 0.90 ± 0.23 |
| | Mean | 1.22 ± 0.28 | 14.78 ± 4.88 | 3562 ± 917 | 0.96 ± 0.26 | 0.97 ± 0.27 |

### 3.2. Two Obvious Peaks Occurred in Tree Height Growth in One Year

According to the height of the various families in each month, the growth trend in height in the *C. axillaris* families was consistent with corresponding groups. As shown in

Figure 2, two obvious growth peaks were observed, from March to May and in August. Meanwhile, slow growth with low values from January to February, June to July, and September to December were also observed. Basically, the growth period of *C. axillaris* tends to last for a relatively short period in a year (Figure 2A). According to the monthly variation of tree height growth, the growth period can be divided into five stages: lag-growing stage, the first fast-growing stage, the first slow-growing stage, the second fast-growing stage, and the second low-growing stage (Table 5).

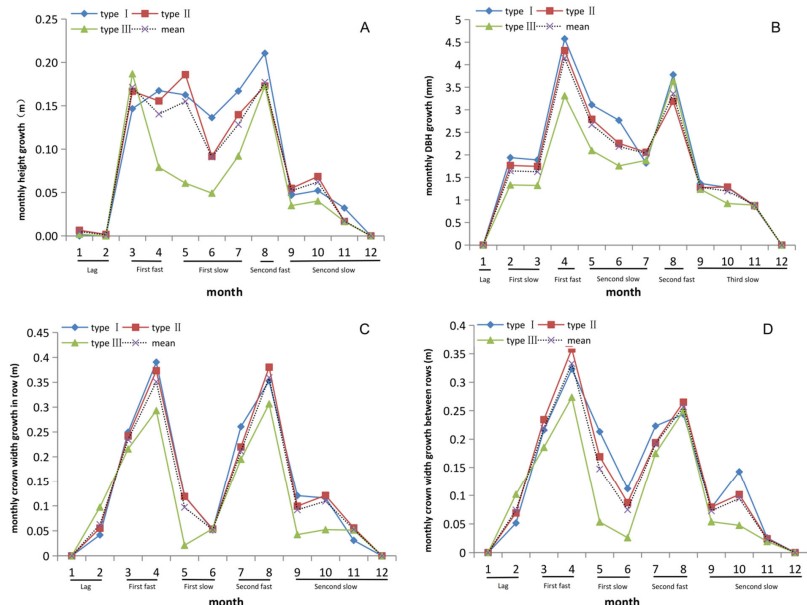

**Figure 2.** Monthly dynamics of growth traits for three types of *C. axillaris* families. (**A**): Monthly dynamics of height growth; (**B**): monthly dynamics of DBH growth; (**C**): monthly dynamics of crown width in row; (**D**): monthly dynamics of crown between rows.

**Table 5.** Height growth stages for three types of *C. axillaris* families.

| Growth Stage | Month | Rapid Growth Type I | Growth Level in One Month (m) Mean Growth Type II | Slow Growth Type III |
|---|---|---|---|---|
| Lag-growing stage | January–February | 0.00 ± 0.00 B | 0.12 ± 0.04 C | 0.08 ± 0.06 D |
| First fast-growing stage | March–May | 0.16 ± 0.11 A | 0.18 ± 0.13 A | 0.13 ± 0.14 B |
| First slow-growing stage | June–July | 0.19 ± 0.14 A | 0.16 ± 0.10 B | 0.10 ± 0.08 BC |
| Second fast-growing stage | August | 0.23 ± 0.18 A | 0.17 ± 0.12 A | 0.17 ± 0.10 A |
| Second slow-growing stage | September–December | 0.08 ± 0.08 B | 0.07 ± 0.06 C | 0.05 ± 0.04 CD |

It was found that group I and II of *C. axillaris* had a similar trend in height growth with type III. Differently, type I and II maintained a more fast-growing stage which was from March to May, while type III happened only in April or August (Figure 2A). Type I, with the largest annual height growth of *C. axillaris*, maintained continuous growth throughout the year. The declining level of growth during the fast-growing period was not as obvious as that in other types, indicating that type I maintained the height growth advantage for a long time. Meanwhile, the growth rate of group II and III in the first fast-growing stage was higher than the growth rate in the second fast-growing stage, while group I showed a faster growth rate in the second fast-growing stage. The fast-growing stage of type III was obviously concentrated in April and August, which was different from that of group I and II.

### 3.3. DBH Stably Increased from February to August in C. axillaris Families

The DBH growth showed significantly differences, although it was not the same as the sharp changes in tree height. Similar with the height of trees, two DBH growth peaks

were observed in *C. axillaris* in April and August, respectively (Figure 2B), which could be divided into six stages: lag stage, the first slow-growth stage, the first fast-growing stage, the second slow-growing stage, the second fast-growing stage, and the third slow-growing stage (Table 6).

**Table 6.** Growth stage divisions of DBH for three growth types in *C. axillaris*.

| Growth Stage | Month | Rapid Growth Type I | Growth Level in One Month (m) Mean Growth Type II | Slow Growth Type III |
|---|---|---|---|---|
| Lag-growing stage | January | 0.00 ± 0.00 E | 0.00 ± 0.00 E | 0.00 ± 0.00 D |
| First slow-growing stage | February–March | 1.91 ± 0.98 CD | 1.78 ± 1.06 C | 1.37 ± 0.64 BC |
| First fast-growing stage | April | 4.58 ± 3.24 A | 4.35 ± 1.97 A | 3.52 ± 1.61 A |
| Second slow-growing stage | May–July | 3.14 ± 1.72 BC | 2.67 ± 1.75 C | 2.17 ± 1.54 B |
| Second fast-growing stage | August | 3.78 ± 2.25 AB | 3.23 ± 1.97 B | 3.63 ± 1.83 A |
| Third slow-growing stage | September–December | 1.20 ± 0.74 DE | 1.28 ± 1.28 D | 1.20 ± 0.95 CD |

DBH entered the first and second fast-growing stages at the same time for the three various types, and the change dynamic was almost the same except for the difference in growth value in three types. As shown in Figure 2B, DBH growth in type I was better than that in type II and III. Meanwhile, a stable growth period happened from February to March, and the growth level of the slow-growth period was from June to July, which was significantly higher than that of the first slow-growth period.

*3.4. Growth Rhythm of Crown Width within Row Showed Differences with Crown Width between Rows*

The growth rhythm of CT showed a similar result with DBH among three types of *C. axillaris* families. Meanwhile, a slight change was observed in each type. Two key growth peaks also occurred in April and August, which was important to the growth of crown width (Figure 2C). According to Figure 1, the growth period of CT can be divided into five stages, i.e., delayed growth stage, the first fast-growing stage, the first slow-growing stage, the second fast-growing stage, and the second slow-growing stage (Table 7). The monthly variation of CT and the start of the growth period was basically the same among the three *C. axillaris* groups. However, CT in type I did not show an obvious advantage. For example, CT of type II was significantly higher than that of other types in the second fast-growing stage. Meanwhile, it was found that type III entered the first slow-growth stage in May, while the other two types entered it in June. After the slow-growth period from September to October, CT growth decreased significantly in the second slow-growth period, with the growth of type I being larger than that of types II and III. However, the growth of type II did not show significant differences in the two fast-growing stages.

**Table 7.** Growth stage divisions of crown width in rows for three types of *C. axillaris* families.

| Growth Stage | Month | Rapid GrowthType I | Growth Level in One Month (m) Mean Growth Type II | Slow Growth Type III |
|---|---|---|---|---|
| Lag-growing stage | January–February | 0.07 ± 0.06 B | 0.08 ± 0.09 C | 0.24 ± 0.34 B |
| First fast-growing stage | March–April | 0.32 ± 0.15 A | 0.31 ± 0.15 A | 0.25 ± 0.13 A |
| First slow-growing stage | May–June | 0.24 ± 0.16 B | 0.18 ± 0.14 B | 0.12 ± 0.10 B |
| Second fast-growing stage | July–August | 0.31 ± 0.20 A | 0.31 ± 0.22 A | 0.25 ± 0.17 A |
| Second slow-growing stage | September–December | 0.12 ± 0.12 B | 0.11 ± 0.11 BC | 0.07 ± 0.07 B |

Differing from CT, an apparent small growth peak occurred in October with similar growth changes among the three types. There were also two high growth peaks in April and August, although the peak value in April was significantly higher than that in August (Figure 2D). The CR growth period can also be divided into five stages: the lag stage, first fast-growing stage, first slow-growing stage, second fast-growing stage, and second slow-growing stage (Table 8). The start time of both fast-growing stages and the overall dynamics was almost the same among three types of *C. axillaris*. For type II, the growth peak in the first fast-growing stage was significantly higher than that in the second fast-growing stage and that of other types in the same fast-growing stage. Moreover, the growth level of type III decreased significantly from May to June.

**Table 8.** Growth stage divisions of crown width between rows for three types of *C. axillaris* families.

| Growth Stage | Month | Growth Level in One Month (m) | | |
|---|---|---|---|---|
| | | Rapid Growth Type I | Average Growth Type II | Slow Growth Type III |
| Lag-growing stage | January–February | 0.07 ± 0.06 C | 0.10 ± 0.10 D | 0.16 ± 0.21 B |
| First fast-growing stage | March–April | 0.27 ± 0.09 A | 0.30 ± 0.14 A | 0.23 ± 0.10 A |
| First slow-growing stage | May–June | 0.27 ± 0.22 B | 0.23 ± 0.18 C | 0.10 ± 0.08 B |
| Second fast-growing stage | July–August | 0.23 ± 0.14 AB | 0.23 ± 0.13 B | 0.21 ± 0.12 A |
| Second slow-growing stage | September–December | 0.12 ± 0.12 C | 0.10 ± 0.11 D | 0.06 ± 0.05 B |

### 3.5. Rainfall and Sunshine Hours Significantly Affect Growth Traits

To identify the effects of meteorological factors in growth traits of *C. axillaris*, Pearson correlation analysis was performed between the growth traits and meteorological factors (Tables 9 and 10). All monthly growth traits were significantly correlated with sunshine hours (SH) (Table 9; $p < 0.10$, 0.05, 0.01), while the increase in V of a single tree in each month was significantly correlated with temperature, rainfall, and humidity (Table 10). Meanwhile, tree height and DBH were also significantly correlated with rainfall and SH, and crown width and CR were highly correlated with SH.

**Table 9.** The correlation analysis between growth traits of *C. axillaris* and meteorological factors of the previous month.

| Factors | Mean Monthly Air Temperature | Mean Monthly Highest Air Temperature | Mean Monthly Lowest Air Temperature | Monthly Rainfall | Monthly Sunny Hours | Relative Air Humidity |
|---|---|---|---|---|---|---|
| Height | 0.2625 | 0.3098 | 0.2473 | 0.1418 | 0.4155 * | 0.1927 |
| DBH | 0.2093 | 0.2618 | 0.1882 | 0.0866 | 0.4018 * | 0.1535 |
| Volume | 0.2830 | 0.3608 | 0.2404 | 0.0282 | 0.5842 *** | −0.0426 |
| Crown width in rows | 0.2940 | 0.3606 | 0.2569 | 0.1140 | 0.5021 ** | 0.0709 |
| Crown width between rows | 0.2966 | 0.3536 | 0.2672 | 0.1190 | 0.4564 ** | 0.1293 |

Note: * significant level at $p < 0.10$; ** $p < 0.05$; *** $p < 0.01$. r (18,0.01) = 0.561, r (18,0.05) = 0.444, r (18,0.10) = 0.378.

**Table 10.** The correlation analysis between monthly growth level of *C. axillaris* and meteorological factors of the previous month.

| Factors | Mean Monthly Air Temperature | Mean Monthly Highest Air Temperature | Mean Monthly Lowest Air Temperature | Monthly Rainfall | Monthly Sunny Hours | Relative Air Humidity |
|---|---|---|---|---|---|---|
| Monthly height growth | 0.1183 | 0.0704 | 0.1167 | 0.4679 ** | −0.4203 * | 0.2619 |
| Monthly DBH growth | 0.3082 | 0.2396 | 0.3269 | 0.6056 *** | −0.3239 | 0.3657 |
| Monthly volume | 0.4521 ** | 0.3942 * | 0.4776 ** | 0.5359 ** | −0.1779 | 0.6901 *** |
| Monthly crown width between tree | −0.2136 | −0.2157 | −0.2282 | −0.0499 | −0.3049 | 0.0211 |
| Monthly crown width between row | −0.1891 | −0.2121 | −0.2017 | −0.0265 | −0.4608 ** | 0.1825 |

Note: * significant level at $p < 0.10$; ** $p < 0.05$; *** $p < 0.01$. r (18,0.01) = 0.561, r (18,0.05) = 0.444, r (18,0.10) = 0.378.

To easily understand the relationship between *C. axillaris* growth rhythm and the monthly dynamics of meteorological factors, the chart of the monthly variation of growth traits and various meteorological factors were plotted. The monthly DBH growth was not strongly correlated with air temperature and air humidity, but closely related to SH, with a significant negative correlation (Figure 3). The monthly dynamics of DBH growth and

rainfall were generally consistent, particularly from January to June, with DBH growth lagged due to rainfall changes in July, and then remaining consistent. The monthly increment of tree height was basically consistent with the temperature, monthly rainfall, SH, and air humidity trends (Figure 4). Monthly height growth showed a different trend to that of monthly rainfall from January to July, and then the two changed in a similar trend. The growth trend in CT and CR was basically the same, i.e., the increment of all types' CT in April was high, the increment in August was high but lower than that in April, and the growth increments from January to February and from September to December were relatively low, i.e., the slow-growth stage (Figures 5 and 6). Only for fast-growing *C. axillaris* (type I), the growth peak of CR in May was higher than that in April. Additionally, the increment between rows in May had a strong relation with values of all meteorological factors during the previous month.

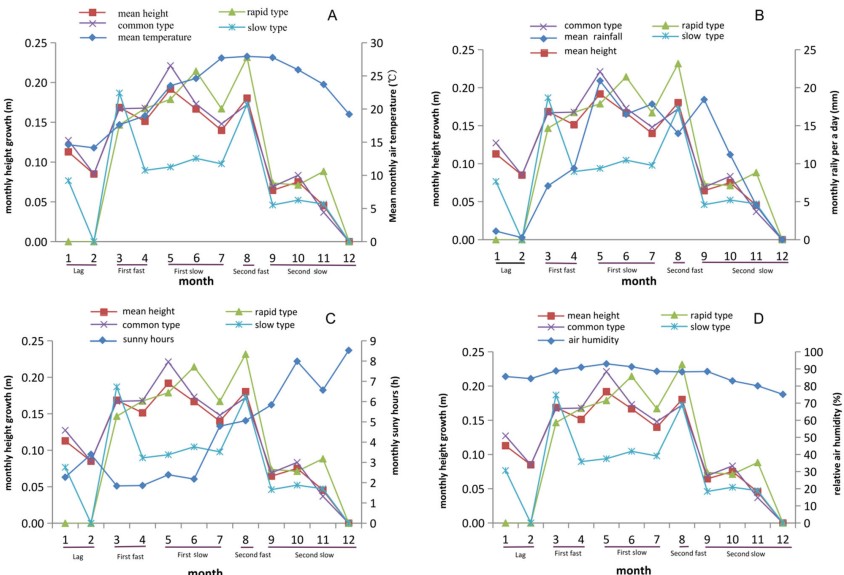

**Figure 3.** Dynamics of monthly height growth and meteorological factors including temperature (**A**); rainfall (**B**); sunshine hours (**C**); and relative humidity (**D**). Meteorological data were supplied by Gaoming Meteorological Bureau, Foshan City, Guangdong Province, China.

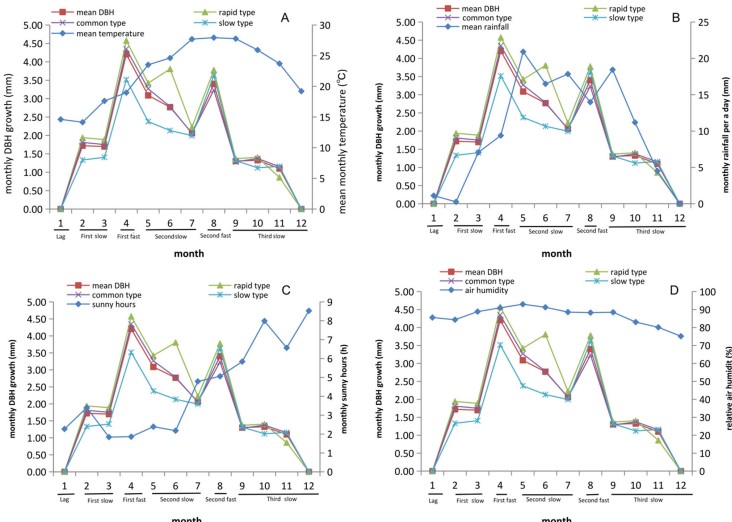

**Figure 4.** Dynamics of monthly DBH growth and meteorological factors including temperature (**A**); rainfall (**B**); sunshine hours (**C**); and relative humidity (**D**). Meteorological data were supplied by Gaoming Meteorological Bureau, Foshan City, Guangdong Province, China.

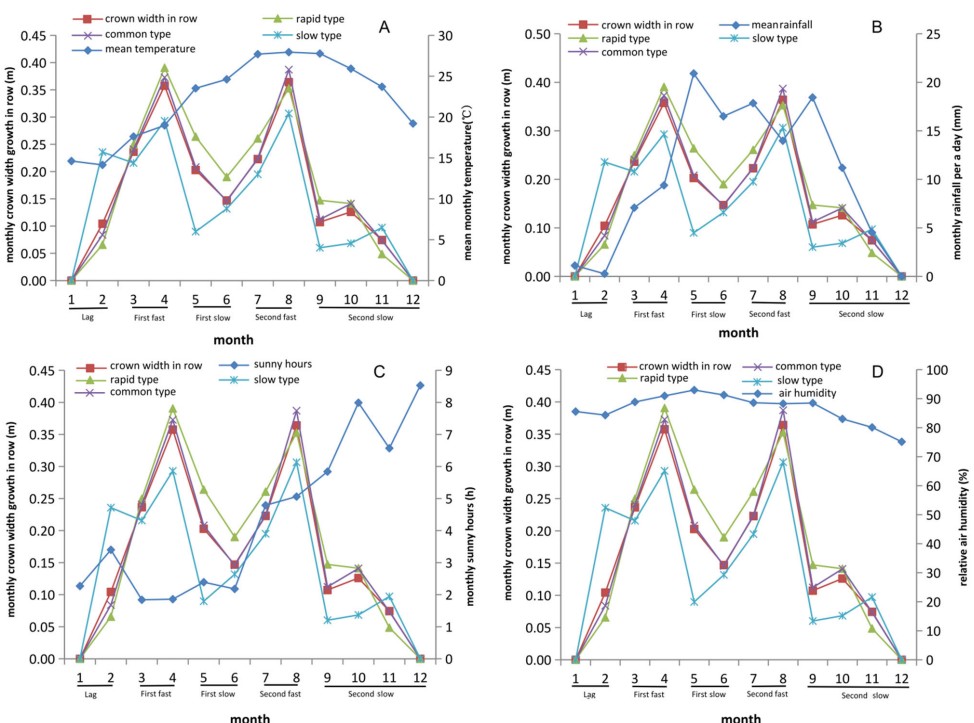

**Figure 5.** Dynamics of monthly crown width in rows and meteorological factors including temperature (**A**); rainfall (**B**); sunshine hours (**C**); and relative humidity (**D**). Meteorological data were supplied by Gaoming Meteorological Bureau, Foshan City, Guangdong Province, China.

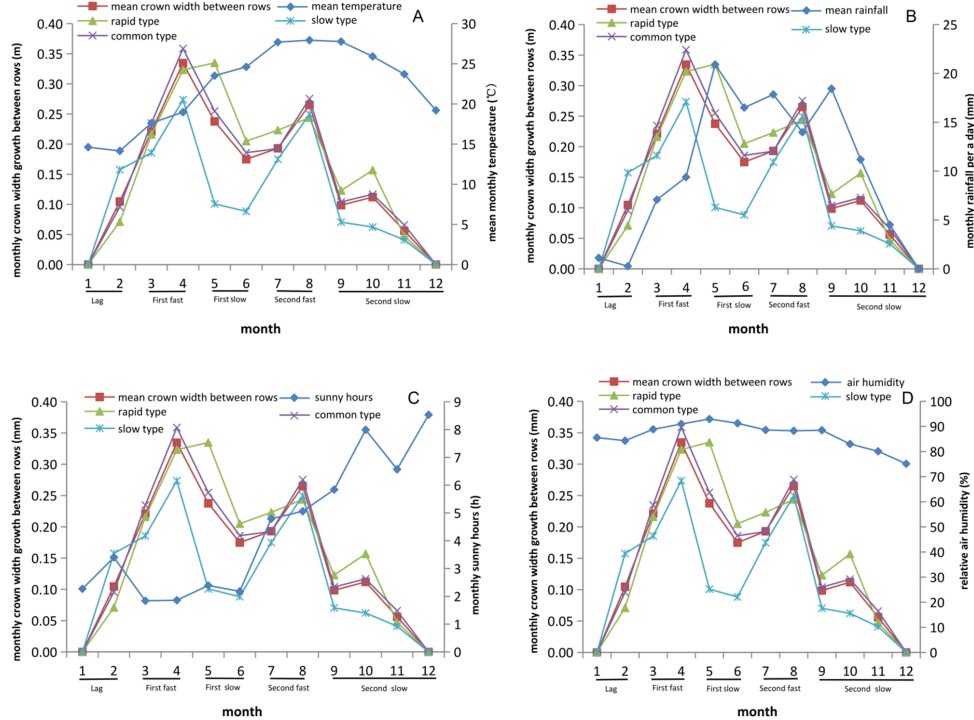

**Figure 6.** Dynamics of monthly crown width between rows and including temperature (**A**); rainfall (**B**); sunshine hours (**C**); and relative humidity (**D**). Meteorological data were supplied by Gaoming Meteorological Bureau, Foshan City, Guangdong Province, China.

## 4. Discussion

*C. axillaris*, as an important tree species used for ecological applications and wood supply, was an essential candidate to develop growth rhythm analysis and selection of superior families. Owing to the lack of systematical and comprehensive research, the development of *C. axillaris* in South China had severely lagged. Hence, it is necessary to develop the research about growth rhythm and further selecting superior families in *C. axillaris*. In this study, 77 families from five provenances were used for selecting superior families and growth rhythm analysis. No. 15, 76, and 56 originated from Raoping provenances, which are located in the east of Guangdong province. Hence, it was concluded that families from Raoping provenances were more suitable for planting in Guangdong province. The suitability of these families in various areas needs to be further verified in the next step of the study.

Tree growth trends reflect the adaptability of its genetic characteristics to environmental conditions. If these are mastered, scientific management can be carried out according to the growth characteristics in different periods of seedling growth to accelerate the growth of trees [38]. *Dendrobenthamia japonica* (DC.) Fang var. chinensis (Osborn.) Fang has a small leaf area and poor stress resistance in the early growth stage, and it requires water and fertilizer management (April to May). Its peak growth period plays a key role in the growth of seedling height and ground diameter. As this period occurs in the summer, i.e., high temperature and drought, it is necessary to strengthen water and fertilizer management [39,40]. In addition, as the growth peak period of ground diameter and seedling height do not coincide, the time of water and fertilizer management in summer should be appropriately extended (June to September); from October to late November, the growth period of *D. japonica* is in the hardening stage, so fertilization should be stopped, and water management should be executed well.

The *C. axillaris* growth rhythm showed that the growth peak was mainly in April and August. The monthly increment of tree height, DBH, and single volume were also significantly correlated with the rainfall of the previous month, and water was the limiting factor for the growth of *C. axillaris*. According to our research, it was obvious that rainfall and fertilizer play an important role in promoting the rapid growth of plants. Recently, a study of the relationships between the growth of *C. axillaris*. and the climate of long-term plantations in East-Central Thailand was performed [41]. Similarly, the growth of *C. axillaris* was positively correlated with moisture, which was the main limiting factor and corresponded with the monsoon season from July to October [41]. In *T. grandis*, a significant correlation between rainfall and the DBH growth of teak also occurred, especially in the rainy season [42]. Contrastingly, there is no obvious correlation between *C. acuminata* and growth in the rainy season and rainfall, and it seems that increasing rainfall in the dry season may have a larger effect on stem radial growth than that in rainy season [42]. Therefore, it is necessary that early artificial forest tending and topdressing are carried out in combination with rainy weather and forest road conditions in rainy months. However, to gain high-quality timber with straight stems and no knots, the slow-growing period from December to February of the following year is needed [43].

Temperature was also one of the main climatic factors affecting plant growth in a southern subtropical climate. Of course, the promoting effect of rainfall on growth may also include the effect of temperature, because temperature is high in the rainy season, and the curve of water heat coefficient (R/T) is basically consistent with that of rainfall. However, the responses of different species to the periodic changes of rainfall and temperature seem to be quite complex [42]. The change of stem growth includes the real growth and the change of trunk volume caused by the change of water content in plants. This volume change caused by changing water content has been reported in the determination of radial growth of trees in tropical forests. This volume change is caused by changing water content and temperature. The growth in the dry season from October to March is generally the smallest, even negative in October, November, and February, while in December and January there is still a certain amount of growth [44,45]. Similar results were also found in *C. axillaris*, in

which the monthly increment of tree height was basically consistent with the temperature. However, the DBH growth was not strongly correlated with air temperature. This may be caused by the decrease in rainfall in October and November, while plant transpiration is still strong, resulting in the rapid decrease in water content in cells and the shrinkage of trunk volume. Meanwhile, the low DBH value in February may be the result of the real deceleration of plant trunk growth under low temperatures in January and February [46].

## 5. Conclusions

The tree height, DBH, single plant volume, crown width between trees and between rows were significantly different among various families ($p < 0.01$). All 77 families were classified into fast-growing type, middle-growth type, and slow-growth type. Meanwhile, the families No. 15, 76, and 56 showed a higher trait of growth than the average values, and were selected as superior families which will be used for propagation and plantation in the future. The growth trend in fast-growing, middle growth, and slow-growing *C. axillaris* (type I, II, and III, respectively) was consistent with the average growth characteristics, although there were differences in the growth rate. Significant differences in tree height and DBH among the three *C. axillaris* types were observed, especially in tree height. All monthly increments of growth traits were significantly correlated with SH, while the monthly growth of single plant volume accumulation was significantly correlated with other meteorological factors. Meanwhile, tree height and canopy width were significantly negatively correlated with SH. Moreover, the monthly increment of growth traits was higher from April to August, and peak values were apparent in April and August. These results will be helpful to future plantation and forest management of *C. axillaris*.

**Author Contributions:** Conceptualization, G.L., J.X. and C.F.; methodology and investigation, G.L., J.L., C.L., H.L., B.M. and M.L.; writing and revision, G.L., J.L. and C.F.; funding acquisition, G.L. All authors have read and agreed to the published version of the manuscript.

**Funding:** Project supported by the Forestry Science and Technology Innovation Project of Guangdong "Research and demonstration on breeding and high efficient cultivation techniques of *Choerospondias axillaris*" (No. 2015KJCX008/2018KJCX021), and by the Technical Standard Project of Guangdong "Technical regulation of seedling cultivation of *Choerospondias axillaris*" (No. 2017-DB-05).

**Acknowledgments:** The authors gratefully thank Huang Xizhao, Ou Jinwei, Li Weimei, Wang Weiqing, Liang Baozhi, and Zhu Yanglin, respectively from the Research Institute of Tropical Forestry and the Gaomin Forestry Research Institute of Foshan for their painstaking assistance in the growth traits survey. Comments from reviewers are also appreciated.

**Conflicts of Interest:** The authors declare no conflict of interest.

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
