# Peer review of "Growth Rhythm Analysis of Young Stand and Selection of Superior Families in Choerospondias axillaris"

_forests, doi:10.3390/f13122145_

Round 1

Reviewer 1 Report

Detailed comments:

Pg.1, Ln 30 – the keywords should be different from the title of the manuscript, I recommend replacing "growth rhythm" and adding some others.

Pg.2, Ln 67-76 – it needs to be reformulated. Part belongs to the methodology. There is a lack of a clear formulation of the goal of the paper and the working hypothesis.

Pg.3, Ln 99 – There is a reference to Table 1 in the text, but it is missing.

Pg.5, Table 3 – the "family codes" column must be formally modified, because some numbers of one family are in two lines (e.g. 16, 39 for Type II and 27 for Type III).

Pg.6, Ln 185 – in the text it is "delayed growth stage" but in Tables 5 to 8 it is "lag-growing stage" - it should be unified.

Pg.7, Ln 198 ; Pg.8, Ln 213, 2014 – really Figure 1?

Pg.15, Ln 403-407 – I recommend removing it. The reader is not interested in the numbers of families, but in general findings.

Author Response

Please check it in the attachment. Many thanks.

Reviewer 2 Report

The objective of the manuscript seems to be the analysis of the growth rhythm in different families of Chroerospondias axillaris young stands. The study was carried out with 77 families of 3 year old. The methods and statistical analysis are simple and correct, but sometimes not very clear and easy to understand. The results and conclusions obtained are based on the findings of the study. Results are not presented very clear.  There is no a clear discussion between the authors’ results and other similar studies. I recommend the publication of the manuscript after the authors make the changes suggested below.

Author Response

(The authors gave the same response as above.)

Round 2

Reviewer 2 Report

The authors have considerably improved the quality of the paper. However, there are some missing things or changes that should be done before publishing it.

Author Response

Dear Editor,

Thank you very much for the opportunity of revision. We here submit our revised manuscript entitled “Growth rhythm analysis of young stand and selection of superior Families in Choerospondias axillaris ”, to be considered for publication in Forests.

We also thank reviewers for their constructive comments and suggestions, which are guidelines for the revisions of our manuscript. We responded questions point-by-point, and summarized the details of response in “response to reviewers. docx”. Correspondingly, all revisions made in the manuscript are highlighted by using “Track Changes” function in “Revised manuscript. docx”, and they can be easily visible for the convenience of your checking. Besides, we also submitted a clean version to make it easy to find the line number.

Thank you very much for considering this manuscript. We are looking forward to hearing from you.

Sincerely,

Chunjie Fan

Research Institute of Tropical Forestry,

Chinese Academy of Forestry,

Guangzhou 510520, People’s Republic of China

GENERAL COMMENTS

The authors have considerably improved the quality of the paper. However, there are some missing things or changes that should be done before publishing it.

Answer: Thanks for your suggestion, we added the related contents and modified according your suggestion.

SPECIFIC COMMENTS

CONCERN 1. INTRODUCTION

- In the previous revision of the manuscript, it was commented that the Introduction section should be improved with more enhanced literature. A better background of the situation of the research in this topic should be done. However, the authors continue without improving this section.

Answer: Sorry for our missing understanding. We added the related literature and reference to improve this section. The details listed in Line 52-62

CONCERN 2. MATERIALS AND METHODS

- Authors continue without explaining how many plots were used in the study. Please, clarify this information.

Answer: Many thanks for this comment. We added the information. 5 plots were analyzed in this study and each plot has 77 families. The details listed on Line110-114.

- In the first review, it was recommended to improve the statistical analysis because they were a bit scarce. The authors have not taken this comment into account and continue to present the same. These are very simple analysis for a scientific article.

Answer: Sorry for our missing understanding. We performed the statistical analysis. Details listed in Table 5-8

- P4 L200. Please include “(Eq. 1)” after the formula to clarify the information in the paragraph.

Answer: We added the “(Eq. 1)” after the formula.

CONCERN 3. RESULTS

- The authors continue without explaining which the growth traits applied in the study were. Please, clarify this information.

Answer: We feel sorry for making you confused. We measured the growth traits including H, DBH, V, CT and CR were measured in this study, which were added in Line 130-131 and Line 150-151.

- P9 l982. Please, change “Figure” for “Figures”. Please, separate “5&” as “5 &”.

Answer: Thanks for your mention, we modified these according your suggestion.

CONCERN 4. DISCUSSION

- This section continues without being improved with more enhanced literature. Authors continue without discussing their results deeply with other results obtained in similar studies. This section should still be improved.

Answer: Thanks for your great suggestion. We added the related literature and developed the discussion of relationship between rainfall and growth. The details listed in Line 299-316, 328-331.

TABLES

- Table 1. Why does only the month of July appear in the year 2017? In 2018, the authors do not present all the months of the year, why? If the information is not available, it should be explained.

Answer: Thanks for your suggestion, we added the information from July, 2017 to December, 2019 in Table 1. During these periods, we performed the investigation.

FIGURES

- It would be advisable to stand out the growth phases in the graphs of the figure to facilitate the understanding of the results. This recommendation was already given in the first review but has not been taken into account by the authors.

Answer: Sorry for our missing understanding. We stand out the growth phrases in all Figures.
